# Withholding of M-CSF Supplement Reprograms Macrophages to M2-Like via Endogenous *CSF-1* Activation

**DOI:** 10.3390/ijms22073532

**Published:** 2021-03-29

**Authors:** Yu-Chih Chen, Yin-Siew Lai, Yan-Der Hsuuw, Ko-Tung Chang

**Affiliations:** 1Graduate Institute of Bioresources, National Pingtung University of Science and Technology, Pingtung 91201, Taiwan; christian--0704@hotmail.com; 2Research Center for Animal Biologics, National Pingtung University of Science and Technology, Pingtung 91201, Taiwan; rindy.0802@gmail.com; 3Department of Tropical Agriculture and International Cooperation, Pingtung 91201, Taiwan; hsuuw@mail.npust.edu.tw; 4Flow Cytometry Center, Precision Instruments Center, National Pingtung University of Science and Technology, Pingtung 91201, Taiwan

**Keywords:** M-CSF, M2-like, macrophage, transcriptome

## Abstract

Macrophage colony-stimulating factor (M-CSF or CSF-1) is known to have a broad range of actions on myeloid cells maturation, including the regulation of macrophage differentiation, proliferation and survival. Macrophages generated by M-CSF stimulus have been proposed to be alternatively activated or M2 phenotype. M-CSF is commonly overexpressed by tumors and is also known to enhance tumor growth and aggressiveness via stimulating pro-tumor activities of tumor-associated macrophages (TAMs). Currently, inhibition of CSF-1/CSF-1R interaction by therapeutic antibody to deplete TAMs and their pro-tumor functions is becoming a prevalent strategy in cancer therapy. However, its antitumor activity shows a limited single-agent effect. Therefore, macrophages in response to M-CSF interruption are pending for further investigation. To achieve this study, bone marrow derived macrophages were generated in vitro by M-CSF stimulation for 7 days and then continuously grown until day 21 in M-CSF absence. A selective pressure for cell survival was initiated after withdrawal of M-CSF. The surviving cells were more prone to M2-like phenotype, even after receiving interleukin-4 (IL-4) stimulation. The transcriptome analysis unveiled that endogenous *CSF-1* level was dramatically up-regulated and numerous genes downstream to *CSF-1* covering tumor necrosis factor (TNF), ras-related protein 1 (Rap1) and phosphatidylinositol 3-kinase (PI3K)-protein kinase B (AKT) signaling pathway were significantly modulated, especially for proliferation, migration and adhesion. Moreover, the phenomenal increase of *miR-21-5p* and genes related to pro-tumor activity were observed in parallel. In summary, withholding of CSF-1/CSF-1R interaction would rather augment than suspend the M-CSF-driven pro-tumor activities of M2 macrophages in a long run.

## 1. Introduction

In the tumor microenvironment (TME), circulating monocytic precursors in blood are recruited to tumor site by the growth factors, such as vascular endothelial growth factor (VEGF), transforming growth factor beta (TGF-β), granulocyte-macrophage colony stimulating factors (GM-CSF) and M-CSF and the chemokines CCL2 and CCL5 [1,2]. Recruited monocytes differentiate into mature macrophages within the TME and become the most abundant immune population in situ. Although early studies demonstrated that optimally stimulated macrophages are capable of killing tumor cells in vitro, it is now generally accepted that tumor-associated macrophages (TAMs) conditioned by TME play a critical role in tumorigenicity [1,2]. Typically, TAMs are designated as “alternatively activated” M2-type macrophages. In tumor, TAMs behave as an accomplice with the immunosuppressive and proangiogenic capacities in the promotion of tumor progression [3,4]. Unlike Th1 and Th2 cells, M1 and M2 macrophages are not stably differentiated subsets and their phenotype can be switched reversibly [4,5].

The pro-tumor roles of TAMs include tumor invasion, tumor angiogenesis, immunosuppression and metastasis [6]. Many biomarkers such as CD68, CD163, CD206, and CD204 have been identified in TAMs. Indeed, researchers have shown that the infiltration of TAMs positively correlates with a poor clinical prognosis in various tumors, such as lung cancer, gastric cancer, gliomas, and lymphomas [4,7,8,9]. In non–small cell lung cancer, the infiltration degree of CD68^+^ TAMs was also associated with the expression of M-CSF in the tumor stroma. The 5-year overall survival rate and disease-free survival rate of patients with histological grade of CD68^−^ M-CSF^−^ were better than their counterpart with CD68^+^ M-CSF^+^ (51.3% vs. 19.6% and 49.3% vs. 15.7%, respectively; *p* = 0.001) [7]. It is consistent with other results that showed significant improvement of progression free survival and overall survival of the patients with low expression of CD206 and CD204 level in glioma tissue and gastric cancer [8,9], respectively.

Infiltration of monocytes/macrophages is often stimulated by tumor derived colony-stimulating factor-1 (CSF-1, also known as M-CSF). CSF-1 not only induces macrophage precursors to mature into adherent motile baseline macrophages (M0), but also regulates macrophages survival, proliferation and differentiation [10,11,12,13]. The effects of CSF-1 are transduced by phosphotyrosine motif-activated signals of its receptor, the CSF-1R [12]. CSF-1R, also known as cFMS, is an exclusive type III receptor tyrosine kinase closely related to tyrosine-protein kinase kit (c-kit) and fms-related tyrosine kinase-3 (flt-3), with high expression limited to macrophage lineage [14,15,16]. In patients with gliomas, a high level of M-CSF expression was always followed by a higher grade glioma. Moreover, the high expression level of M-CSF was also significantly correlated with the ratio of CD163 to CD68 on TAMs in grade IV gliomas. These data indicated that polarization of macrophages towards the M2 phenotype correlated with higher histological grade of gliomas secreting M-CSF [17].

Immunotherapy acts in a fundamentally different way in comparison with conventional therapies. Instead of destroying tumor cells directly, immunotherapy directs tumor cell killing through the immune response of the host. This mission can be achieved mainly via the effector cells of the immune system, such as macrophages [4]. Therefore, TAMs has been becoming a promising therapeutic target for cancer therapy. Strategies to deplete TAMs by inhibition of CSF-1/CSF-1R signaling required for macrophages survival is now considered a promising therapeutic approach in the clinic [4]. Kubota et al. reported that daily anti-CSF-1R therapeutic antibody AFS98 treatment (50 mg/kg) reduced the number of peritumoral LYVE-1^+^ macrophages in an implanted osteosarcoma (AX cells) mice model and reduced vascularization, lymphangiogenesis and tumor growth [18]. An orally active CSF-1R kinase inhibitor (BLZ945, 200 mg/kg), which readily passes through blood brain barrier, has been shown to specifically target macrophages depletion, enhance CD8^+^ T cell infiltration, and prevent tumor growth in the K14-HPV16 transgenic mouse model of cervical carcinoma [19]. BLZ945 was again being proved to reduce macrophages and improve survival and malignancy in the glioma mice model [4,20].

Human monocyte-derived TAMs were able to be generated in vitro while culturing in conditioned media collected either from human breast adenocarcinoma cell or LK 46 lymphoblastoid cell [21]. The expression of *miR-21* was induced by CSF-1R signaling via PI3K activation [22]. In an animal study, inhibition of *miR-21* in TAMs induced a proinflammatory angiostatic function of macrophage with improvement of immunostimulatory response [23]. Meanwhile, *miR-21* blockade in TAMs also increases CD107/CD8 positive T-cells in TME. All these effects led to diminished tumor growth [23].

Now a day, inhibition of CSF-1/CSF-1R interaction by therapeutic antibody to deplete TAMs and their pro-tumor functions is becoming a prevalent strategy in cancer therapy. Although inhibition of macrophages infiltration in tumor-bearing mice via CSF-1R blockade had a significant effect on tumor regression and survival of CD8^+^ T cells [24,25,26]. However, its antitumor activity that ranges from showing no efficacy to no objective response or limited single-agent effect were frequently reported in solid tumor clinical trials [27,28,29]. A phase Ib/II trial with the combination of Program Death-1 (PD-1) and CSF-1R inhibitor was thereby strongly suggested in patients with advanced solid tumor [29]. In O’Brien, et al.’ study, treatment of antibody against mouse CSF1R (clone M279) on day 0 right after tumor cells implantation resulted in greater tumor growth inhibition compared to that on day 12. This outcome was interpreted by the authors that intervention of antibody blockade in late phase of tumor development could differentially affect the distribution of anti- and pro-tumorigenic TAMs [30]. Furthermore, Zhu et al. found that CD206^Hi^ TAMs were more sensitive to the CSF1R signal blockade (clone 5A1) bringing a higher level of cell death. They also showed that CSF1R blockade reprograms survival CD206^Low^ TAMs to support anti-tumor interferon responses and T cell activities [26]. Thereby, macrophages in response to post-M-CSF stimulation are pending for further investigation. Hence, removal of CSF-1/CSF-1R interaction by replacement of fresh media to withhold M-CSF signaling is our approach to see the post-M-CSF reaction in macrophages.

## 2. Results

### 2.1. Withholding of M-CSF Supplement Still Directs M2 Macrophages Polarization for Long Time

To test the effect of M-CSF long-lastingly in macrophages, bone marrow mononuclear cells were stimulated with M-CSF for 7 days, then the cells were continuously grown without supplement of M-CSF from day 7 till day 21 (Figure 1A). By day 7 with M-CSF stimulation, 90% of total cells expressed CD11b as a recognition of bone marrow derived monocytes/macrophages (BMDMs) (Figure 1B). The expression of CD206 as a marker of macrophage M2-type was significantly higher on cells on day 7 (P3) and day 21 (P4) compared with those on day 0 (P2) (Figure 1C,D). Fluorescent immunocytochemistry (ICC) staining confirmed that higher proportion of cells co-expressing CD11b and CD206 from P3 and P4 fractions were compared with those in P2 (Figure 1E). The cells from P3 and P4 fractions revealed a spindle-like morphology for M2-type macrophages (Figure 1E, Bright Field). It is in parallel with the study from Weiss et al. (2013) showing elongated fibroblast-like shaped cells in a majority of M-CSF-mediated BMDMs [31].

### 2.2. Sustainable Expression of CD206 on Single Cell after Withholding of M-CSF Supplement

The mean expression of CD206 in a cell population represented by the mean fluorescent intensity (MFI) were maintained from day 7 till day 21 (Figure 2A,C). Besides, the MFI of CD86 showed no significant difference in parallel (Figure 2B,D). The expression of CD206 versus CD86 were 1.5 ± 0.15 to 2.04 ± 0.51 (Figure 2E).

### 2.3. A Selective Pressure for Cell Survival Was Initiated after Withholding of M-CSF Supplement

With M-CSF supplement, BMDMs showed high proliferation rate. However, total cell counts were decreased significantly after withdrawal of M-CSF, but surprisingly they maintained equally in number between day 14 and day 21 (Figure 3A). Cell shrinkage, blebbing and exposure of phosphatidylserine were observed as a phenomenon of programmed cell death (Figure 3B,C). We also confirmed a high level of early apoptotic cells (Annexin V^+^/7-AAD^−^) starting from day 7 till day 21 (Figure 3D). Taken together, these data indicated that withholding of M-CSF supplement caused a cell stress for BMDMs and a regimen for cell survival was also initiated.

### 2.4. Trend to Polarization into M2, Instead of M1 Subtype, after Withholding of M-CSF Supplement

The tumor microenvironment (TME) contains a variety of cytokines and growth factors other than M-CSF. We were interesting in the cross-reaction of M-CSF with other stimuli, such as lipopolysaccharides (LPS), IL-4 or IL-10, on M2-like macrophages induction from cells collected from P3 and P4 fractions. Interestingly, the gene transcript of interferon regulatory factor 1 and 5 (IRF1 and IRF5), known as M1 biomarkers, were significantly downregulated in P4 (D21) compared with that in P3 (D7) macrophages while LPS was further added (Figure 4A,B). On the contrary, the gene transcripts of arginase 1 (ARG1), known as M2 biomarker, were significantly upregulated both in P3 and P4 macrophages upon further IL-4 stimulation (Figure 4C). We therefore further examined the gene expression upon IL-4 stimulus in a broad survey by transcriptome analysis; and unexpectedly, more than 10 genes associated with the regulation of M2 phenotype were expressed significantly higher in P4 macrophages than that in P3 (Figure 4E). Meanwhile, we also found that more than 10 genes associated with M1 phenotype were expressed significantly lower in P4 macrophages than P3 vice versa (Figure 4D). Gene regulation in macrophages for M1- and M2-associated phenotypes were previously described [32,33,34,35,36,37,38,39].

However, there were no significant changes between P3 and P4 macrophages in the reaction with further IL-10 stimulus. Since IL-4 and IL-10 can direct M2-like macrophages polarization, these results manifested that cells from P4 in comparison with P3 fraction were not only more resistant to re-polarization into M1 subtype, but also more preferential to go toward M2-like polarization in response to further LPS or IL-4 stimuli, respectively. These findings also attract a major concern that cytokines milieu within TME may enhance M2-like TAMs development upon M-CSF interruption. The underlying biological and molecular mechanisms for how macrophages from P4 vs. P3 fractions behaving so differently after interruption of M-CSF supplement were explored further via transcriptome profiling.

### 2.5. Endogenous Expression of CSF-1 Was Augmented in M2 Macrophages after Withholding of M-CSF Supplement

We next compared the transcriptome profiling of M2 macrophages collected from P3 and P4 fractions by Next Generation Sequencing (NGS). Differential expression analysis (DEA) revealed 272 differentially expressed genes (DEGs) in P4 compared with P3 (152 upregulated and 120 downregulated; false discovery rate [FDR] ≤ 0.05, Figure 5A). Histograms for the top 20 upregulated and downregulated genes overall and for each transition are shown (Figure 5B,C). M2 macrophages from P4 fractions, with abundant expression of extracellular matrix associated cellular component gene (COL1A1, COL1A2, COL3A1, COL5A1, COL5A2, COL12A1, FN1, TNC, FBLN2, CTGF, SPARC, BGN, TIMP3, HSPG2), adhesion associated gene (CSF1), angiogenesis associated gene (THBS2), actin filament organization associated gene (FAT1), smooth muscle associated gene (CALD1), glucose transmembrane transport associated gene (RSC1A1) and metabolic process associated biological process gene (CEMIP) compared to that from P3 fractions. In contrast to the preponderance of extracellular matrix related makers in the upregulated gene set, the top downregulated genes in P4 macrophages encoded proteins with disparate functions, including biological process (CLEC4A1, ERMARD), glycolytic process (ENO1B), metabolic process (DYNLT1B, ATP6V0C-PS2, FUNDC2), cell cycle process (CCNA2, PAGR1A, CDK20, HJURP, DLGAP5), mRNA processing (SRSF7), cellular response (IFI209), response to virus (OAS1A, EVI2), defense response (CCR5, HIST1H2BK) and unknown (BCC003331, CYP4F16). Consistent with previous study [40], we found that the top upregulated Gene Ontology (GO) pathways regulated cell motility, cell adhesion and cell migration mostly (Figure 5D) and the top downregulated GO pathways regulated metabolism, cell cycle and cellular response, and so forth. (Figure 5E).

Unexpectedly, the expression of endogenous M-CSF (also called CSF-1) tremendously upregulated in cells from P4 compared with that from P3 fraction (Figure 6A). Rap1 signaling pathway, PI3K-Akt signaling pathway and TNF signaling pathway were unveiled from the Kyoto Encyclopedia of Genes and Genomes (KEGG) pathway enrichments of differentially expressed genes (DEGs) closely related with CSF-1 expression in P4 macrophages (Figure 6B,C). Besides, a number of gene transcripts such as CREB3L1, PTGS2, THBS1, PDGFRB, PDGFC, FPR1, ADORA2A, TLN2, RAPGEF3, FN1, TNC, COL1A1, COL1A2, COL4A1, COL4A2, COL6A1, COL6A3, THBS2, LAMB1, CCND2, LAMA5 were upregulated in cells from P4 compared with that in P3; on the contrary, CCL5, TRAF1, CASP7, TIAM1, BRCA1 and CCNE1 were downregulated in P4 versus P3 (Figure 6B). Moreover, there were 16 shared genes of any two or three pathways in the overlapping areas of the Venn diagram. Those genes were associated with the pro-tumor activity (Figure 6C). GO analysis highlighted a number of enriched terms, such as response to external stimulus, cell motility, cell migration, cell locomotion, chemotaxis and cell adhesion (FDR ≤ 0.05) (Figure 6D). The interferon regulatory factor 7 (IRF7), its increased expression associated with M2 to M1 switch in microglia in vitro and in vivo [41], showed significantly decreased expression in P4 compared with P3 macrophages (Figure 6E). Meanwhile, DEA unveiled that some enriched terms, such as regulation of defense (GO) and cytokine production (GO), Epstein-Barr virus infection (KEGG) and viral carcinogenesis (KEGG) were closely related to IRF7 expression (FDR ≤ 0.05) (Figure 6F).

### 2.6. Upregulation of the Genes Related to Pro-Tumor Activity in M2-Like Macrophages after Withholding of M-CSF

A previous study reported that the expression of miR-21 was induced by CSF-1R signaling via PI3K pathway activation [42]. Comparing with our results, we also showed the phenomenal increases of miR-21-5p expression both in macrophages from P3 and P4 fractions (Figure 7A) while the PI3K-Akt pathway was consistently activated (Figure 6B). Given that the media was changed every 2 days in our protocol, it is reasonable to assume that the cultures were not completely deprived but supported instead with an autocrine signaling of M-CSF. According to KEGG pathway enrichment of selected DEGs, the upregulation of pro-tumor activities in M2 macrophages after interruption of M-CSF supplement were highlighted. Those pro-tumor functions for M2 macrophages included angiogenesis, ECM-receptor interaction, collagen digestion and absorption and cancer cell metastasis (Figure 7B).

## 3. Discussion

The proliferation of BMDMs can be flexibly regulated by changing the concentration of growth factor M-CSF [43,44]. Our results showed in advance a sustainable expression of CD206 on CD11b macrophages after supplemental interruption of M-CSF for 14 days (Figure 1C,F, and Figure 2E). Although withholding of M-CSF supplement caused a cell stress for BMDMs and pushed them to go toward early apoptosis (Figure 3), a regimen for cell survival, mainly directed by tremendously higher expression of endogenous *M-CSF*, was also initiated (Figure 6A). After M-CSF withdrawal, macrophages predominantly had resistance to M1 re-polarization by further LPS stimulation and were more prone to M2a or M2-like polarization upon additional IL-4 stimulus (Figure 4). Although the expression of *ARG1* was attenuated in P4 compared with that in P3, it still showed a high expression compared with the control group (~200 folds), very likely due to a consistent M-CSF autocrine signaling. These findings attract a major concern that Th2 cytokines milieu within TME containing IL-4 may enhance M2-like TAMs development, especially while the infiltrating macrophages interact with M-CSF intermittently by therapeutic CSF-1/CSF-1R antibody blockade. According to global transcriptome analysis in our study (Figure 5), numerous genes downstream to *CSF-1* were significantly modulated in macrophages after supplemental interruption of M-CSF. They are majorly covered by TNF, Rap1 and PI3K-Akt signaling pathway for the activation of pro-tumor function (Figure 6B,C). To further validate our findings, macrophages from P3 versus P4 were analyzed in GO pathways. Consistent with the previous report that showed the development of an adherent phenotype of BMDMs upon exposure to increasing doses of M-CSF [40], the GO pathways from our data also manifested the cells from P4 versus P3 fractions in the regulation of cell motility, cell adhesion and cell migration (Figure 6D). Those higher activities indicates faithfully the M2-like phenotypes of macrophages.

The network cross-linkage of TNF, Rap1 and PI3K-Akt signaling pathway were frequently reported in the regulation of cell survival, proliferation and migration. For example, tumor necrosis factor-α (TNF-α), an extraordinarily pleiotropic cytokine, is indispensable from a central role of immune homeostasis, inflammation, and host defense [45]. Increased blood levels of TNF-α have been associated in metastatic cancer patients [46]. Furthermore, the higher expression of TNF receptor 1 also can promote proliferation of C4HD murine mammary tumor cells in vitro and in vivo through activation of the p42/p44 MAPK, JNK, PI3K-Akt pathways [47,48]. TNF-α also increases the proliferation of growth-competent mouse BMDMs in the presence of M-CSF [49,50]. The mRNA expression and protein levels of Rap1 were increased in THP-1 macrophages in a time-dependent manner after TNF-α stimulation [51]. In terms of Ras-associated protein-1 (Rap1), it is a small GTPase in the Ras-related protein family and plays important roles in the regulation of tumor cell invasion and metastasis [52,53]. Rap1 can bind to either guanosine triphosphate (GTP) or guanosine diphosphate (GDP). When associated with GTP, Rap1 binds to specific Rac guanine nucleotide exchange factors (GEFs), including VAV2 and TIAM1, to activate Hela cell mobilization [54,55]. In addition, Rap1 controls cell survival and proliferation via activation of phosphatidylinositol-3-kinase (PI3K) pathway [55]. Phosphorylation of CSF-1R Y721 also mediates its association with PI3K-Akt pathway to regulate BMDMs motility [56]. The inhibition of PI3K-Akt by a specific inhibitor BEZ235 can significantly suppress cell proliferation, migration, and invasion of HT-29 and HCT-116 cells [57]. Besides, treatment of another PI3K inhibitor, LY294002, could suppress both the transcriptional and protein expression of TNF-α in RAW264.7 cells with post-stimulation of ceramic and titanium particles [58].

The *microRNA-21 (miR-21)* transcriptional regulation was demonstrated firstly by Loffler et al., who showed that *miR-21* gene is induced by IL-6 in a STAT3-dependent event [59,60]. Giving support to this, they identified two STAT3 binding sites in the putative *miR-21* promoter. The cumulative evidences suggest that *miR-21* is an oncogene conserved in vertebrates locating in 17q23.2 [61]. Chan et al. demonstrated that aberrant expression of *miR-21* in cultured glioblastoma cells leads to the malignant phenotype by blocking expression of critical apoptosis-related genes [62]. Another xenograft mouse model also demonstrated that *miR-21* is able to promote tumor growth [63]. In recent study, the diagnostic values of *miR-21* were verified, mainly by the positive correlation of *miR-21* and M-CSF expression in cervical cancer (*r* = 0.6825, *p* < 0.001) [64]. Another report also manifested that the expression of *miR-21* was induced by CSF-1R signaling via PI3K pathway activation [42]. According to our result, it had shown that the expression level of *miR-21* was equally high in cells between P3 and P4 fractions after withholding of M-CSF. In contrast, the expression level of *miR-21* was significantly lower in M-CSF pretreated control (P2) (Figure 7A). This result acts in cooperation with our prior data showing a much higher expression of endogenous *CSF-1* in cells after withdrawal of M-CSF (Figure 6A); suggesting that *miR-21* level was maintained by an uncutted CSF-1R signaling. Besides, our data are also in parallel with previous study demonstrating that the expression of *miR-21* in M2 macrophages was significantly higher than in M0 and M1 macrophages [65].

## 4. Materials and Methods

### 4.1. Animals

C57BL/6 female mice were purchased from the National Laboratory Animal Center (Taipei, Taiwan) and housed in a clean and pathogen-free facility in the National Pingtung University of Science and Technology animal care facility. Eight to twelve weeks old mice were used in the experiments. The protocol was approved by the Institutional Animal Care and Use Committee of College (IACUC) of Veterinary Medicine at National Pingtung University of Science and Technology (IACUC Approval No.: NPUST-108-072, 24 April 2020), Taiwan. Euthanasia was performed by cervical dislocation after anesthesia.

### 4.2. Differentiation of Bone Marrow-Derived Macrophages (BMDMs)

Bone marrow cells were obtained by flushing the marrow cavities of the femurs and tibias with phosphate buffered saline (PBS) (pH 7.4), containing 0.5% bovine serum albumin (BSA) (Sigma-Aldrich, Steinheim, Germany). Bone Marrow-derived Mononuclear cells (BMMNCs) were isolated from the buffy coat by Ficoll-Hypaque density gradient centrifugation (GE Healthcare, Uppsala, Sweden). BMMNCs were differentiated into macrophages by incubating in RPMI 1640 (Corning, Manassas, VA, USA) with supplemented 10% fetal bovine serum (PAA, Pasching, Austria), 50 ng/mL M-CSF (#315-02, Peprotech, Rocky Hill, NJ, USA) at 37 °C in 5% CO_2_ for 7 days. After 7 days, Bone Marrow-derived Macrophages (BMDMs) were harvest by using 0.025% Trypsin–EDTA (Hyclone, Logan, UT, USA) and or continuing cultured for 21 days at 37 °C in 5% CO_2_ in 90% RPMI 1640 (Corning, Manassas, VA, USA) with supplemented 10% fetal calf serum (PAA, Pasching, Austria) and changed every 2–3 day until 21 days.

### 4.3. Macrophage Subtype Polarization

On day 7 in culture, the BMDM cells were washed, counted and re-plated in 2 mL RPMI 1640 media (Corning, Manassas, VA, USA) containing with 10% fetal bovine serum (Hyclone, Logan, UT, USA) at a density of 5 × 10^4^ cells/ cm^2^ (Cellstar 6-well plate) (Greiner Bio-One, Kremsmünster, Austria). Cells were classically activated (M1 phenotype) with 1 μg/mL LPS (Sigma-Aldrich, Steinheim, Germany), and alternatively activated (M2 phenotype) with 10 ng/ ml IL-4 (CELL guidance system, St. Louis, MO, USA) or with 10 ng/mL IL-10 (Peprotech, Rocky Hill, NJ, USA) for 24 h at 37 °C and 5% CO_2_.

### 4.4. Immunocytochemistry

Fluorescent immunocytochemistry (ICC) staining of CD11b and CD206 were performed on BMDMs and mounted on Cover Slips. CD11b^-^ and CD11b^+^ cells were separated by using CD11b MicroBeads (Miltenyi Biotec, Bergisch Gladbach, Germany). The cells were first fixed in 4% paraformaldehyde solution for 15 min at room temperature, followed by blocking with 1% BSA in PBST solution (Phosphate-buffered saline with 0.1% Tween^®^ 20 detergent) for 30 min. Cells were then incubated with Rat biotin anti-mouse CD11b antibody (M1/70) (Biolegend, San Diego, CA, USA) and Rabbit anti-mouse CD206 polyclonal antibody (Elabscience, Houston, TX, USA) for 1 h before the application of a secondary antibody (anti-rat Streptavidin- FITC (Biolegend, San Diego, CA, USA), Goat anti-rabbit IgG CF594 (Biotium, Fremont, CA, USA)) for 1 h. Slides were stained with DAPI before imaging. Images were captured with a fluorescence microscope (IX71, Olympus, Tokyo, Japan).

### 4.5. Flow Cytometry Analysis

BMDM cells were collected at various time points and were analyzed by flow cytometry for M1 or M2 cell- lineage markers. The cells were stained in PBS containing 0.5% (wt/vol) BSA (Sigma-Aldrich, Steinheim, Germany). Surface receptors were stained for 30 min on ice. The identification of M1 type in BMDM was performed with Fluorescein isothiocyanate (FITC)- conjugated CD86 antibody (GL-1) (1 μg/mL). The identification of M2 type in BMDM was performed with APC-conjugated CD206 antibody (C068C2) (0.5 μg/ mL). All antibodies were obtained from Biolegend (San Diego, CA, USA). We use unstained cells as a negative control and single-stained cells as a compensation control.

### 4.6. Apoptosis Assay

BMDM cells were analyzed at various time points (Day 7, 14, and 21) by flow cytometry and fluorescent immunocytochemistry assay. The cells were then trypsinized and washed with PBS twice. Apoptosis was measured by flow cytometry using the FITC Annexin V Apoptosis Detection Kit with propidium iodide (Biolegend, San Diego, CA, USA). For fluorescent immunocytochemistry assay, the cells were re-seeded into culture plates (5 × 10^4^ cells/cm^2^) and allow cells to adhere, followed by adding 10 µL/mL of the pSIVA-IANBD probe (Bio-Rad, Hercules, CA, USA) on cells. The cells were observed under microscope using the green fluorescence filter.

### 4.7. Real-Time PCR

The total RNA was isolated from cells using Tripure Isolation Reagent (Roche Diagnostics GmbH, Mannheim, Germany) following manufacture’s instruction. Reverse transcription was performed using iScript cDNA Synthesis Kit (#1708891, Bio-Rad, Hercules, CA, USA) for cDNA synthesis and the quantitation of gene expression was achieved from Luna^®^ Universal quantitative Polymerase Chain Reaction (qPCR) Master Mix (#M3003, New England Biolabs, Ipswich, MA, USA) following manufacture protocol. Quantitative PCR reactions were performed using a QIAGEN Rotor Gene Q Real-Time PCR. The primer sequences (5′–3′; forward, reverse) used in qPCR was showed in Table A1. Amplification was achieved for 40–50 cycle following enzyme activation step in 95 °C for 1 min hold, denaturation in 95 °C for 15 s, and extension step in 60 °C for 30 s. The expression of target genes was normalized to internal control β-actin. The fluorescence signal was detected at the end of each cycle. Melting curve analysis was used to confirm the specificity of the products. The data were calculated using the comparative methods of the −ΔC*t* (-delta Ct) or 2^−ΔΔC*t*^ (standardized mRNA level).

### 4.8. Library Preparation and Sequencing

The purified RNA was used for the preparation of the sequencing library by TruSeq Stranded mRNA Library Prep Kit (Illumina, San Diego, CA, USA) following the manufacturer’s recommendations. Briefly, mRNA was purified from total RNA (1 μg) by oligo (dT)-coupled magnetic beads and fragmented into small pieces under elevated temperature. The first-strand cDNA was synthesized using reverse transcriptase and random primers. After the generation of double-strand cDNA and adenylation on 3′ ends of DNA fragments, the adaptors were ligated and purified with AMPure XP system (Beckman Coulter, Beverly, MA, USA). The quality of the libraries was assessed on the Agilent Bioanalyzer 2100 system and a Real-Time PCR system. The qualified libraries were then sequenced on an Illumina NovaSeq 6000 platform with 150 bp paired-end reads generated by Genomics, BioSci & Tech Co., New Taipei City, Taiwan. The datasets presented in this study can be found in online repositories (NCBI BioProject PRJNA665327: https://dataview.ncbi.nlm.nih.gov/object/PRJNA665327?reviewer=o2ppa2v2g4db9pu1t2mh3il4i9, accessed on 24 September 2020).

### 4.9. Bioinformatics Analysis

The bases with low quality and sequences from adapters in raw data were removed using program Trimmomatic (version 0.39) [66]. The filtered reads were aligned to the reference genomes using Bowtie2 (version 2.3.4.1) [67]. A user-friendly software RSEM (version 1.2.28) was applied for the quantification of the transcript abundance [68]. Differentially expressed genes (DEGs) were identified by EBSeq (version 1.16.0) [69]. The functional enrichment analysis of Gene Ontology (GO) terms and Kyoto Encyclopedia of Genes and Genomes (KEGG) pathways among gene clusters was implemented in an R package called clusterProfiler (version 3.6.0) [70,71,72].

### 4.10. MicroRNA Expression Analysis

The cDNAs were produced from total RNA by reverse transcription using the miRCURY LNA miRNA PCR Starter Kit (Cat no. 339320, QIAGEN, Germantown, MD, USA). The miRNA PCR primer set, based on the SYBR Green miRCURY locked nucleic acids (LNA) detection system from QIAGEN, was used to profile the expression of *miR-21-5p* (Cat no. YP00204230, QIAGEN, Germantown, MD, USA). The expression level of *UniSp6* (U6) was used as an endogenous control according to the manufacturer’s instructions.

### 4.11. Statistical Analysis

Data are expressed as mean ± SD. Statistical significance of differences among three or more groups were determined using One-way analysis of variance (ANOVA) or two-way ANOVA with Tukey’s multiple comparisons test. *p* < 0.05 was considered statistically significant. The analysis was performed using GraphPad Prism 7 (GraphPad Software, Inc., San Diego, CA, USA).

## 5. Conclusions

Taking together all the results, withholding of the M-CSF supplement dramatically augments pro-tumor activity of M2 macrophages through endogenous *CSF-1* activation and its downstream genes reprogramming (Figure 6 and Figure 7). In the future, immunohistochemistry staining of CSF-1 on TAMs from paraffin embedded tumor tissue should be addressed. Although macrophages from in vitro culture may not fully represent in vivo tissue-resident counterpart, our data still provide useful information and imply that physical withholding of CSF-1/CSF-1R interaction may rather enhance than suspend M2-like macrophages development in TME. Thereby, we could explain the possible limitation of CSF-1R inhibitor in solid tumor treatments. The depletion of TAMs by the intervention of TNF, Rap1 and PI3K-Akt signaling pathway may become a determined approach according to our results. Nevertheless, the delivery of *miRNA-26a* to suppress endogenous *CSF-1* upregulation in cells after CSF-1/CSF-1R axis blockade may apparently be another option for combined therapy [73].

## Figures and Tables

**Figure 1 ijms-22-03532-f001:**
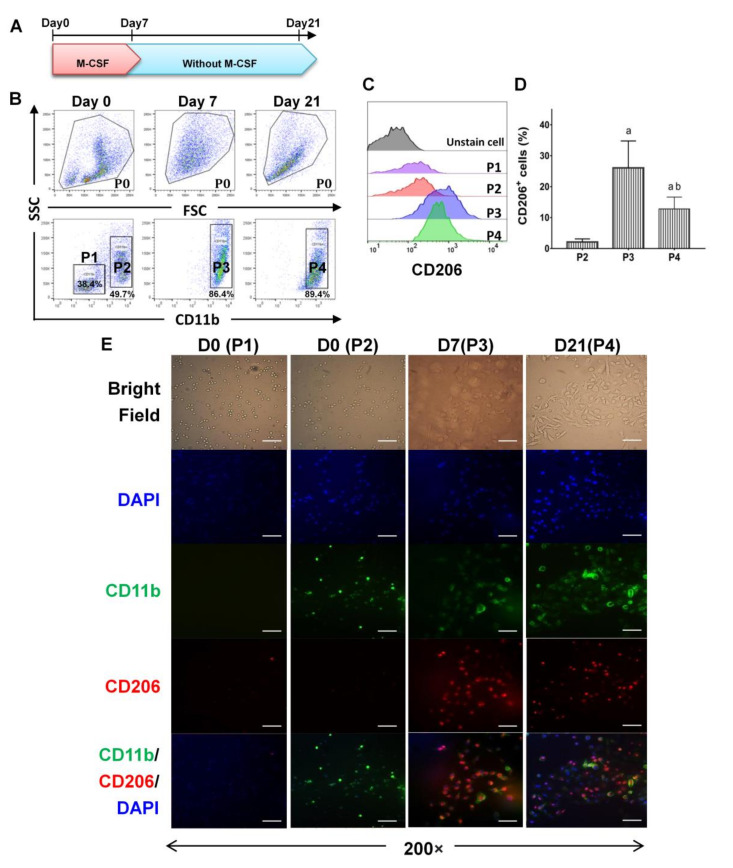
The expression of CD206 was maintained predominantly on CD11b^+^ macrophages after withholding of macrophage colony-stimulating factor (M-CSF) supplement. (**A**) The mononuclear cells collected from murine bone marrow were differentiated into macrophages in vitro in the presence of 50 ng/mL of M-CSF for 7 days. In the end of day 7, media were replaced by a fresh media (RPMI 1640 with 10% of fetal bovine serum) without M-CSF supplement then the cells were cultured continuously for additional 14 days. (**B**) Flow cytometry profiles of CD11b^+^ monocytes/macrophages on day 0, day 7 and day 21, representatively. (**C**,**D**) The expression of CD206 in CD11b^+^ cells were determined by flow cytometry. Data were analyzed using FlowJo V10 software and statistical calculation was presented as mean ± SD, n = 6. a, *p* < 0.05 when compared with P2; b, *p* < 0.05 when compared with P3. (**E**) Representative data of immunocytochemistry of bone marrow derived macrophages treated with M-CSF through day 0 to 7, and after supplemental interruption till day 21. Cells were stained with CD11b (green) and CD206 (red) and the nuclei were stained with 4′,6-diamidino-2-phenylindole (DAPI) (blue). Scale bar is 50 um (magnification 200×).

**Figure 2 ijms-22-03532-f002:**
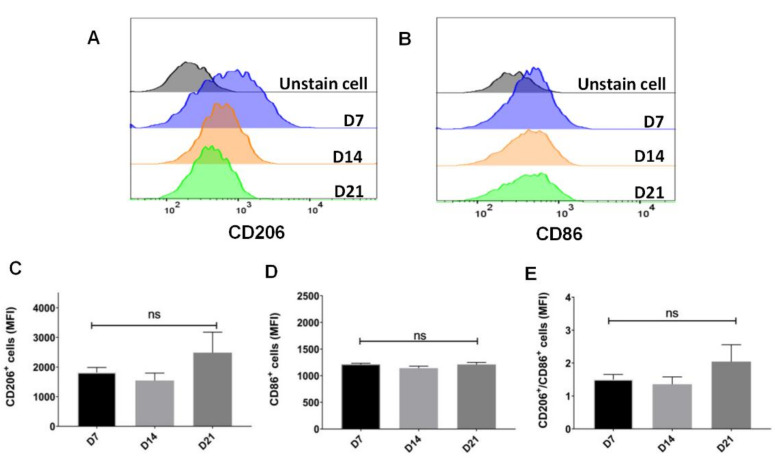
The CD206 expression of single cell on average after withholding of M-CSF supplement. (**A**,**B**) Flow cytometry analysis of mean fluorescence intensity of CD86 and CD206 as M1 and M2 markers, respectively, on total macrophages treated with M-CSF (D7) and after supplemental withdrawal (D14 and D21). (**C**,**D**) Statistics of mean fluorescent intensity of CD86 and CD206 surface makers on total macrophages. (**E**) Statistics of mean fluorescent intensity of CD206/CD86 ratio on total macrophages. Data are mean ± SD (*n* = 3).

**Figure 3 ijms-22-03532-f003:**
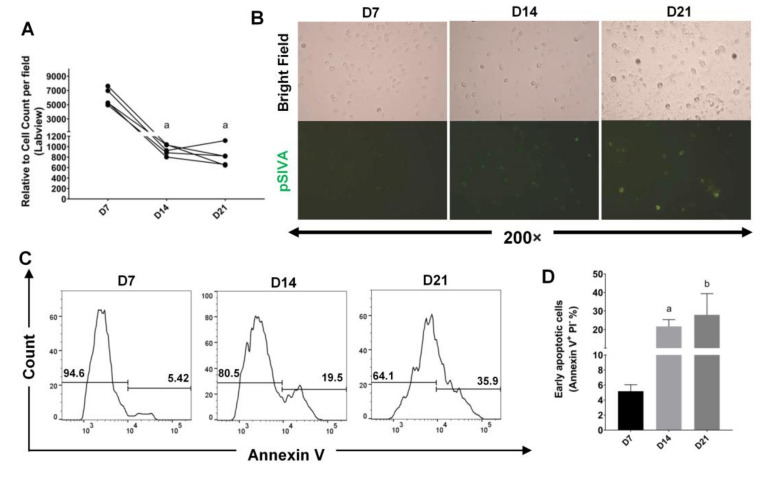
The CD11b^+^ cells underwent early apoptosis after withholding of M-CSF. Supplemental withdrawal of M-CSF elevated early apoptosis in CD11b^+^ macrophages on day 14 (D14) and day 21 (D21) compared with that on day 7 (D7). (**A**) The cell counts of macrophages on D7, D14 and D21 were determined under the microscope, and the digital images were analyzed by Labview (*n* = 5). (**B**) The apoptotic cells were observed under the fluorescent microscope by pSIVA staining (original magnification: 10 × 20). (**C**,**D**) Flow cytometry analysis of apoptotic cells with the expression of FITC-Annexin V on macrophages. Data were mean ± SD (*n* = 3). a, *p* < 0.05 when compared with D7; b, *p* < 0.05 when compared with D14.

**Figure 4 ijms-22-03532-f004:**
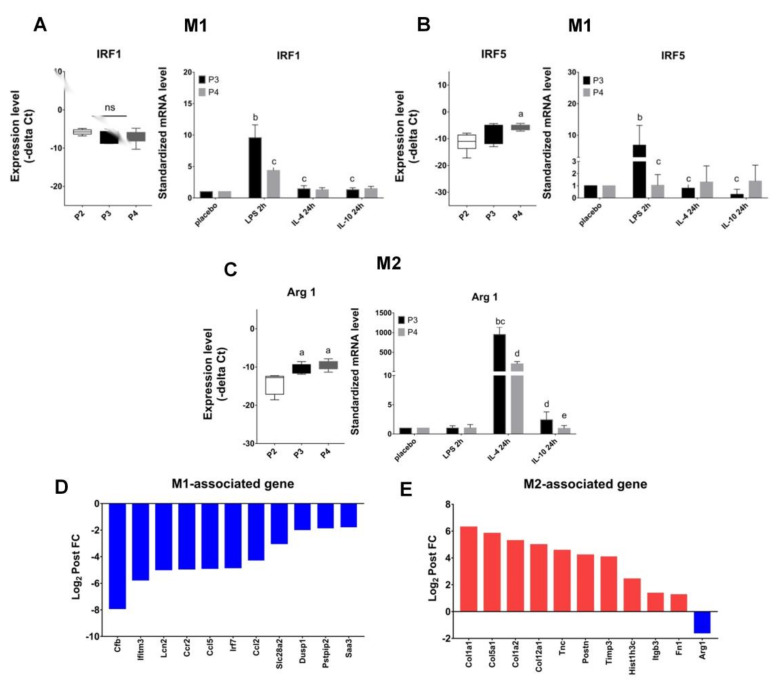
Trend for macrophages to M1/M2 polarization by further LPS, IL-4 or IL-10 stimuli. Macrophages collected from day 7 (P3) and day 21 (P4) culture were further stimulated with either LPS, IL-4 or IL-10, respectively. (**A**–**C**) The expression of M1 macrophages associated genes (*IRF1* and *IRF5*) and M2 macrophages associated genes (*ARG1*) were observed. Data were mean ± SD, *n* = 6. (**D**) Histograms of the M1 macrophages associated genes downregulated by further IL-4 stimulus in day 21 (P4) fraction compared with that in day 7 (P3), *n* = 5, FDR ≤ 0.05. (**E**) Histograms of the M2 macrophages associated genes upregulated by further IL-4 stimulus in day 21 (P4) fraction compared with that in day 7 (P3), *n* = 5, the false discovery rate (FDR) ≤ 0.05. The asterisks indicate statistical significance. a, *p* < 0.05 when compared with P2; b, *p* < 0.05 when compared with P3 placebo group; c, *p* < 0.05 when compared with P3 plus LPS group; d, *p* < 0.05 when compared with P3 plus IL-4 group; e, *p* < 0.05 when compared with P4 plus IL-4 group; ns: no significant difference.

**Figure 5 ijms-22-03532-f005:**
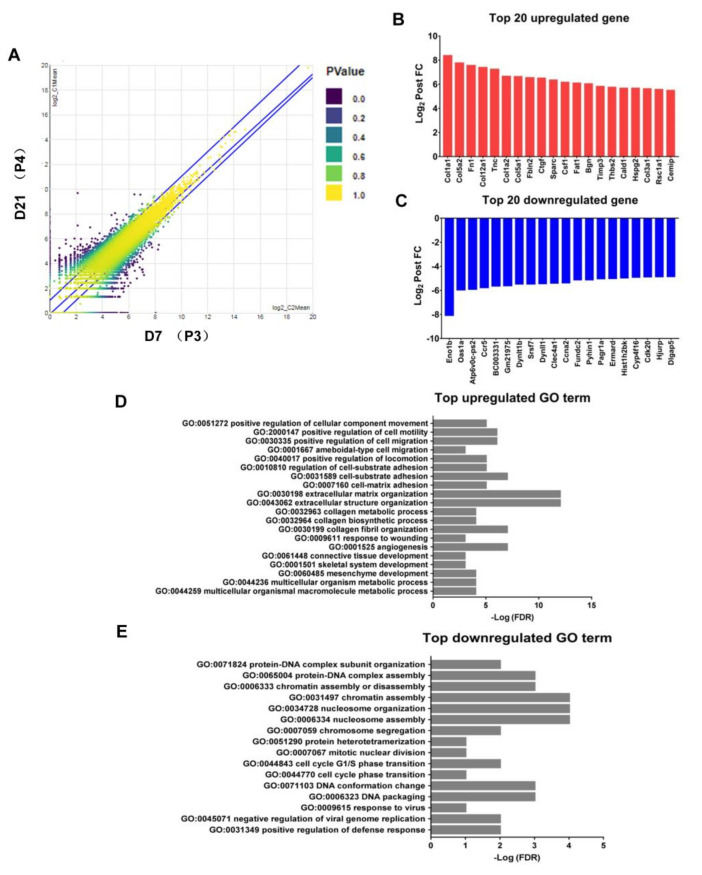
Transcriptome analysis of Top 20 differentially expressed genes in P4 vs. P3 M2 macrophages. (**A**) The scatter plot revealed 272 differentially expressed genes (DEGs) in macrophages on day 21 (P4) compared with that on day 7 (P3). (**B**,**C**) Histograms of the top 20 upregulated genes (red) and downregulated genes (blue). (**D**) Bar plot of the top upregulated biological process gene ontology (GO) analysis in P4 compared with P3. (**E**) Bar plot of the top downregulated biological process GO analysis in P4 compared with P3 (FDR ≤ 0.05), *n* = 5.

**Figure 6 ijms-22-03532-f006:**
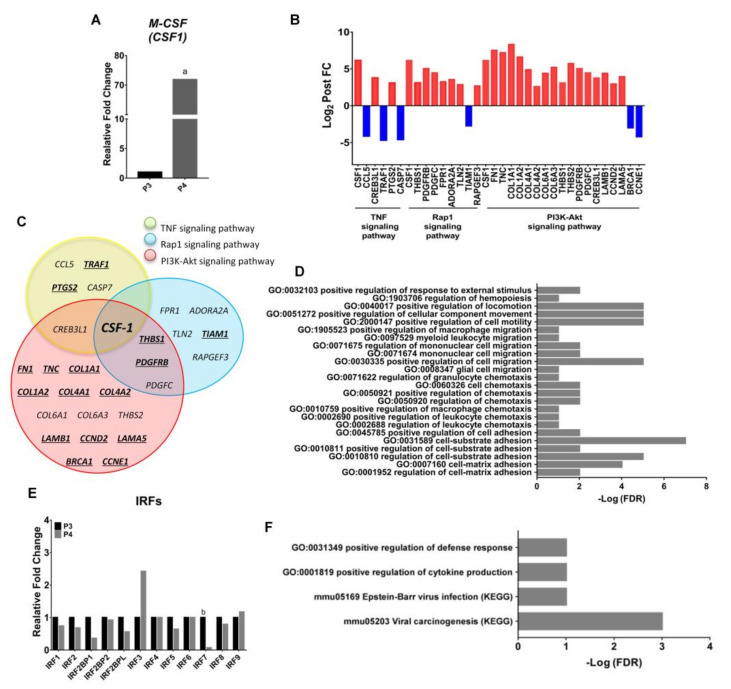
Transcriptome analysis of M2 macrophages from P4 modulated by endogenous *CSF-1* reactivation and gene reprograming. (**A**) Fold change of *M-CSF* (*CSF-1*) in P4 compared to P3. (**B**) KEGG pathway enrichments of selected DEGs in P4 compared with P3 (blue: downregulated genes; red: upregulated genes). (**C**) Venn diagram of three KEGG pathways were highly associated with endogenous *CSF-1* expression. The yellow circle represents the DEGs in TNF pathway, the blue circle represents the DEGs in Rap1 pathway and the red circle represents the DEGs in PI3k-Akt pathway. The overlapping areas indicate the shared genes of any two or three groups. (**D**) Bar plot of selected DEGs were related with *CSF-1* expression in P4 compared with P3. (**E**) Expression of IRFs family gene in P4 compared with P3. (**F**) Bar plot of selected DEGs were related with *IRF7* expression in P4 compared with P3. *n* = 5, a, *p* < 0.05 when compared with P3; b, *p* < 0.05 when compared with P4.

**Figure 7 ijms-22-03532-f007:**
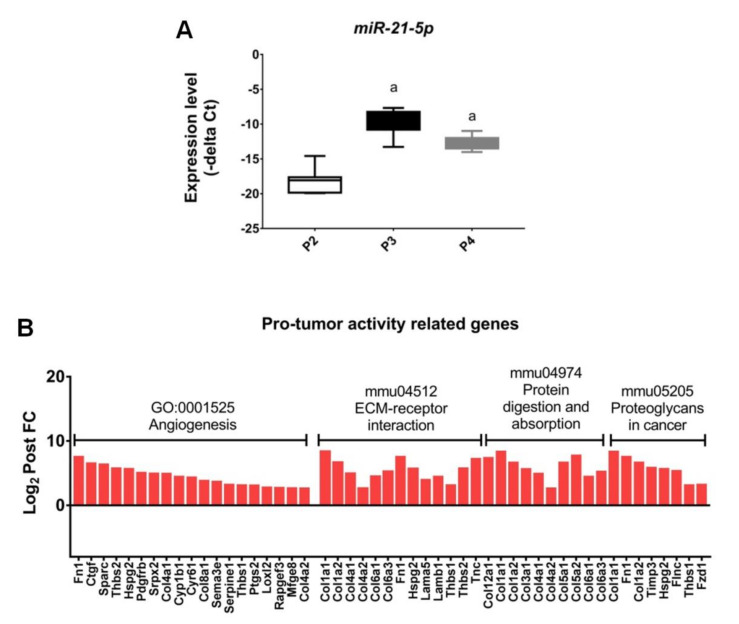
Upregulation of the related genes for pro-tumor activity in M2 macrophages from P4 fraction. (**A**) The expression of *miR-21-5p* in M2 macrophages was measured on day 0 (P2), day 7 (P3) and day 21 (P4), respectively. Data were mean ± SD, *n* = 7. The asterisks indicate statistical significance. a, p < 0.05 when compared with P2. (**B**) KEGG pathway enrichments of selected DEGs for pro-tumor activities in P4 fraction compared with that in P3 (red: upregulated genes). (FDR ≤ 0.05), *n* = 5.

## Data Availability

Not applicable.

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
