# Peer review of "Withholding of M-CSF Supplement Reprograms Macrophages to M2-Like via Endogenous CSF-1 Activation"

_ijms, 2021, doi:10.3390/ijms22073532_

Round 1

Reviewer 1 Report

The question driving this research concerns how macrophages might respond to therapeutic CSF-1 receptor blockade during anti-cancer treatment.  The concern arises from the potential for CSF-1 blockade may result in unintended alterations of macrophage populations into pro-tumor phenotypes.  The authors used an in vitro model to test the effect of CSF-1 deprivation on macrophage polarization and gene expression in bone marrow derived macrophages that consisted of expanding BMDMs for 7 days in the presence of CSF-1 followed by withdrawal of the cytokine for an additional 21 days.  This was to mimic the effects of receptor blockade of signaling with the presumption that treatment acts to impede CSF-1 driven signaling in vivo. The study showed cells cultured under these reduced CSF-1 levels adopted a more M2 phenotype that was resistant to trans-differentiation into a more inflammatory, anti-tumor M1-like phenotype when challenged with LPS.  They show differences in gene expression that may produce a more tumor promoting state. However, these data fail to support a model where therapeutic manipulation of the CSF-1 receptor pathway and result in the unintended consequence of producing more resilience within the tumor microenvironment due to increased immunosuppressive effects within the myeloid compartment.  It is unclear whether the results are due to manipulations similar to what would be expected on CSF-1 signaling or is rather an in vitro artifact within a population that is likely to be dissimilar to tumor resident or infiltrating myeloid cells.  The study also ends at the level of screening for potential markers of effects that might explain therapeutic resistance.  If they had applied the results of this screen and found similar alterations in gene expression in treated vs untreated tumors the results would be more compelling.

Minor points:

The English in the paper needs to be improved for clarity.

The monoclonal antibody clones used for FACS experiments should be identified.

Line 374, Manassas is a city, change to Virginia to maintain parallel with other references. Steinheim is in Germany.  Most locations need to be fixed, follow the format used for biolegend.

On line 142 the mean should be stated followed by standard deviation rather than the range.

Major points:

  1. While the approach followed here is justified by the cost of therapeutic agents, it is not clear from the current literature what the role of ADCC in the removal of macrophages is in the therapies the authors seek to relate their results to. ADCC of TAMs would be expected to result in the production of inflammatory mediators that will influence the trafficking and phenotypes of immune cells within the tumor bed, including monocyte lineage cells. This may impact the phenotype of TAMs in more significant and determinative ways than the withdrawal of CSF-1.
  2. The authors claim that their approach was necessitated by the expense associated with Emactuzumab, however, a ligand-blocking monoclonal antibody, clone AFS98, directed against the murine c-fms receptor is available from BioXcell. Either PLX5622 or BLZ945 c-Fms inhibitors could also be used to eliminate signaling to test the hypothesis.
  3. The authors deprive the cultures of CSF-1 and report a concomitant increase in de novo CSF-1 production in the surviving cells. Given that the media was changed every 2 days, it is reasonable to assume that the cultures were not CSF-1 deprived but supported instead by autocrine signaling. This defeats the central thesis that the model was based on to mimic biologics reduction in CSF-1 signaling.   This raises the significant question of whether cells were driven to an altered phenotype or selected from a subpopulation.  Either way, no evidence is provided that treatment of tumors with anti-CSFR antibodies or inhibitors result in the altered expression of any of the genes identified in the authors system.
  4. There is also no patient data to suggest a real correlation with therapy which could be accomplished for example by IHC or RNAScope staining of histopathology specimens analyzed for the differences gene expression observed in the model system, for example, CSF-1 staining in surviving macrophages.

Author Response

Manuscript # IJMS-1122524

The authors thank the reviewers for spending considerable time and efforts on examining this manuscript and greatly appreciate the constructive comments and the suggestions made by the reviewers. The authors have tried to respond to all the comments and accommodate most of the suggestions in the revised manuscript.

The authors’ answer:

        The authors thank and appreciate the reviewer’s comments. Therefore, we’ve revised our manuscript according to reviewer advises.

Minor points:

The English in the paper needs to be improved for clarity.

The monoclonal antibody clones used for FACS experiments should be identified.

Line 374, Manassas is a city, change to Virginia to maintain parallel with other references. Steinheim is in Germany. Most locations need to be fixed, follow the format used for biolegend.

On line 142 the mean should be stated followed by standard deviation rather than the range.

Minor points answer:

        The authors thank to the reviewer for this attentive reminder. We have done English proofreading and editing again. Meanwhile, the affiliations of companies are fixed and some modifications suggested by Reviewer#1 are highlighted in yellow color in the manuscript. On line 142, the statement is revised by the reviewer’s suggestion highlighted in yellow color.

Major points 1:

While the approach followed here is justified by the cost of therapeutic agents, it is not clear from the current literature what the role of ADCC in the removal of macrophages is in the therapies the authors seek to relate their results to. ADCC of TAMs would be expected to result in the production of inflammatory mediators that will influence the trafficking and phenotypes of immune cells within the tumor bed, including monocyte lineage cells. This may impact the phenotype of TAMs in more significant and determinative ways than the withdrawal of CSF-1.

Major points 1 answer:

        The authors thank to this suggestion raised by the reviewer. The role of ADCC in the removal of macrophages is stated in “introduction”, page 2 and reference [18]. While Zhu et al. targeted CSF1R signaling using the αCSF1 neutralizing antibody (clone 5A1) in murine tumor-bearing mice, the quantification of CD206Hi and CD206Low TAM subsets revealed that αCSF1 treatment for 8 days led to a >90% depletion of CD206Hi TAMs, while CD206Low TAMs decreased by only ~45%. The loss of CD206Hi TAMs could result from either preferential killing of this TAM subset or altered CD206 expression. To distinguish between these possibilities, they analyzed the kinetics of macrophage cell death and found that in PDAC tumors, CD206Hi TAMs experienced significantly higher levels of cell death following αCSF1 treatment than CD206Low TAMs. These data suggest that CD206Hi TAMs are more sensitive to the CSF1R signal blockade. Consistent with this differential sensitivity, we found that CD206Hi TAMs express higher levels of CSF1R. Despite extensive loss of macrophages, 40–50% of TAMs remain after αCSF1 treatment. To determine whether CSF1 blockade reprograms the remaining macrophages to support anti-tumor activities, they FACS sorted TAMs from 8-day vehicle or αCSF1-treated mice bearing established KI tumors and compared their gene expression profiles. TAMs from αCSF1-treated tumors displayed reduced expression of immunosuppressive molecules, including Pdcd1lg2Il10Arg1Tgfb1and Ccl22. By contrast, anti-tumor immunity genes, such as Il12aIfna, Ifnb1, Ifng, Cxcl10, and Nos2, were upregulated. they also observed markedly increased surface expression of MHCII after CSF1 or CSF1R inhibition. Taken together, these data suggest that the CSF1/CSF1R blockade reprograms remaining TAMs to support anti-tumor interferon responses and T cell activities’

        Therefore, we added some statement as highlighted in green color in according to the reviewer’s comment.

Major points 2:

The authors claim that their approach was necessitated by the expense associated with Emactuzumab, however, a ligand-blocking monoclonal antibody, clone AFS98, directed against the murine c-fms receptor is available from BioXcell. Either PLX5622 or BLZ945 c-Fms inhibitors could also be used to eliminate signaling to test the hypothesis.

 Major points 2 answer:

        An antibody against mouse CSF1R was done by O’Brien, et al. In their study, mice were treated with αCSF1R clone (M279) or control antibody starting on the day of tumor implantation (day 0) or on day 12 when tumor’s size up to ~ 100mm3 and continually treated 3 times per week until day 20. Interestingly, they observed that initiating treatment at day 0 resulted in greater tumor growth inhibition compared to day 12.  In response to the reviewer’s comment, we added “O’Brien, et al’s results” at the end of the introduction as highlighted in yellow color for our study’s rationale.

        Meanwhile, we omitted our inappropriate statement “For instance, Afuco™ (Emactuzumab) ADCC enhanced anti-CSF-1R antibodies (CAT#: AFC-TAB-H23) costs 10,000 USD for 1 mg“from the introduction too.

Major points 3:

The authors deprive the cultures of CSF-1 and report a concomitant increase in de novo CSF-1 production in the surviving cells. Given that the media was changed every 2 days, it is reasonable to assume that the cultures were not CSF-1 deprived but supported instead by autocrine signaling. This defeats the central thesis that the model was based on to mimic biologics reduction in CSF-1 signaling. This raises the significant question of whether cells were driven to an altered phenotype or selected from a subpopulation. Either way, no evidence is provided that treatment of tumors with anti-CSFR antibodies or inhibitors result in the altered expression of any of the genes identified in the authors system.

Major points 3 answer:

        The authors agree with the reviewer’s suggestion. It is reasonable to assume that the fresh media exited M-CSF autocrine. However, the concentration of M-CSF was at least maintained in a minimal level during media refreshment for every 2 days. In our unpublished data, it also showed no significant differences in M-CSF concentration in cell culture during day 7- day 21. Thus, our protocol is actually in accordance with the central thesis that the model was based on to mimic biologics reduction in CSF-1 signaling. Hence, we added the statement suggested by the reviewer’s comment to our Result 2.6 as highlighted in yellow color.

        In addition, the deprivation of M-CSF caused a cell stress for BMDMs and pushed the surviving cells being selected to go toward an alternative M2-like phenotype, mainly directed by tremendously higher expression of endogenous M-CSF and its downstream PI3K signaling pathway, etc. Although our result may not be formulated completely in vivo, but it still can provide some important information to us. Hence, we added a statement in Conclusion as highlighted in yellow color.

Major points 4:

There is also no patient data to suggest a real correlation with therapy which could be accomplished for example by IHC or RNAScope staining of histopathology specimens analyzed for the differences gene expression observed in the model system, for example, CSF-1 staining in surviving macrophages.

Major points 4 answer:

The authors appreciate the reviewer’s comment. We therefore add a statement in Conclusion as highlighted in yellow color.

Reviewer 2 Report

Chen et al describe an interesting observation that withholding GM-CSF treatment in BMDM culture can push macrophages to M2.  They indicate that this may have impact on treatments that target CSF1R.

Overall the manuscript suggest an interesting observation that may explain discrepancies in treatment efficacies between patients and cancer types.  However, the following comments may improve the content of the manuscript:

1. The title does not clearly convey the key message of the article and could use revising.  Overall, grammar and English language requires revising.

2. The introduction requires revising, especially regarding the rationale for the study.  Although in the discussion they state that the M2-driven macrophages without GM-CSF is an important observation that may impact therapies that specifically target this axis, the introduction indicates that the importance of this study is a cost-basis matter.  The former has more significance and importance and should be revised accordingly.

3. The authors use "in consistent" (line 59 and throughout manuscript).  I believe they mean just "consistent". Ex: The data is consistent; which would mean agreeing. While inconsistent would indicate disagreeing data.  Please check and revise accordingly.

4. Apoptosis was measured in the cells treated with GM-CSF and those withheld from GM-CSF.  The results indicated that the lack of GM-CSF stimuli pushed cells into stress, and promoting apoptosis thus lowering cellular viability.  No such experiment was performed with the LPS and IL-4 study.  Could cellular viability and apoptosis been affected?

5. Arg1 expression is associated with immunosuppresive subsets of myeloid cells in the TME.  The authors indicate that M2 macrophages grown under IL-4 without GM-CSF drives M2 genes, however an attenuated increase in Arg1 is seen.  The authors should include in the discussion their interpretations and speculations about this observation and its implications in the TME and potential treatments.

6. The authors suggest that this observation could impact treatments that target CSF1R in cancers as even without GM-CSF, macrophages were able to display a M2 phenotype and IL-4 strengthened this change.  The authors should add in the discussion how they think this could affect therapy and why therapies that target CSF1R may be effective or ineffective in certain types of cancers.  Please see PMID: 33511454 DOI: 10.1007/s00262-021-02861-3

7. The methodology for flow cytometry doesn't indicate if controls were used (single color, unstained, FMO, compensation beads etc) and should be included.

8. The methodology for qRT-PCR indicates that the delta-delta Ct method was used for comparative analysis, however your graphs show delta-CT.

Author Response

Manuscript # IJMS-1122524

The authors thank the reviewers for spending considerable time and efforts on examining this manuscript and greatly appreciate the constructive comments and the suggestions made by the reviewers. The authors have tried to respond to all the comments and accommodate most of the suggestions in the revised manuscript.

The authors’ answer:

        The authors thank and appreciate the reviewer’s comments. Therefore, we’ve revised our manuscript according to reviewer advises.

1. The title does not clearly convey the key message of the article and could use revising.  Overall, grammar and English language requires revising.

Answer:

The authors thank to reviewer’s reminder. We have done English proofreading and the title has changed according to the reviewer’s suggestion.

2. The introduction requires revising, especially regarding the rationale for the study.  Although in the discussion they state that the M2-driven macrophages without GM-CSF is an important observation that may impact therapies that specifically target this axis, the introduction indicates that the importance of this study is a cost-basis matter.  The former has more significance and importance and should be revised accordingly.

Answer:

The authors appreciate the reviewer’s comment. In response to the reviewer’s comment, we added some statements according to the reviewer’s suggestion in the Introduction as highlighted in yellow color.

3. The authors use "in consistent" (line 59 and throughout manuscript).  I believe they mean just "consistent". Ex: The data is consistent; which would mean agreeing. While inconsistent would indicate disagreeing data.  Please check and revise accordingly.

Answer:

The authors thank to reviewer for this precious reminder. We have checked and revised the words.

4. Apoptosis was measured in the cells treated with GM-CSF and those withheld from GM-CSF.  The results indicated that the lack of GM-CSF stimuli pushed cells into stress, and promoting apoptosis thus lowering cellular viability.  No such experiment was performed with the LPS and IL-4 study.  Could cellular viability and apoptosis been affected?

Answer:

Apoptosis assay was done for the explanation why the number of macrophages dramatically reduced right after the deprivation of M-CSF supplement in Result 2.3. While using LPS or IL-4 in our experiment was to test the tendency of those surviving macrophages about their M1 or M2 repolarization.  

5. Arg1 expression is associated with immunosuppressive subsets of myeloid cells in the TME.  The authors indicate that M2 macrophages grown under IL-4 without GM-CSF drives M2 genes, however, an attenuated increase in Arg1 is seen.  The authors should include in the discussion their interpretations and speculations about this observation and its implications in the TME and potential treatments.

 Answer:

The authors thank to reviewer’s suggestion. The expression of Arg1 was still high (~200 fold) in macrophages after M-CSF deprivation compared with that in control group. The attenuated expression of Arg1 in P4 compared with that in P3 might be due to a minimal concentration of M-CSF autocrine after media refreshment for every 2 days. Therefore, we added the interpretation of this result in the Discussion as highlighted in yellow color.

6. The authors suggest that this observation could impact treatments that target CSF1R in cancers as even without GM-CSF, macrophages were able to display a M2 phenotype and IL-4 strengthened this change.  The authors should add in the discussion how they think this could affect therapy and why therapies that target CSF1R may be effective or ineffective in certain types of cancers.  Please see PMID: 33511454 DOI: 10.1007/s00262-021-02861-3

Answer:

The authors thank to the reviewer for providing a very useful reference paper. According to O’Brien et al study, we added some statement in Introduction as well as in the Conclusions as highlighted in yellow color.

7. The methodology for flow cytometry doesn't indicate if controls were used (single color, unstained, FMO, compensation beads etc) and should be included.

Answer:

The authors thank and appreciate the reviewer’s reminder. We have corrected the statement.  In response to the reviewer’s comment, we added “We use unstained cells as a negative control and single‐stained cells as a compensation control. ” in Material and Methods 4.5 (Page 14).

8. The methodology for qRT-PCR indicates that the delta-delta Ct method was used for comparative analysis, however, your graphs show delta-CT.

Answer:

The authors thank and appreciate the reviewer’s reminder. We have corrected the statement.  In response to the reviewer’s comment, we added “−△Ct (-delta Ct) or 2-△△Ct (standardized mRNA level)” in Material and Methods 4.7 (Page 14).

Round 2

Reviewer 2 Report

The revisions made are acceptable.